# Remote-carbonyl-directed sequential Heck/isomerization/C(sp²)−H arylation of alkenes for modular synthesis of stereodefined tetrasubstituted olefins

Runze Luan[1,2,3], Ping Lin[1,2], Kun Li[1,2,4], Yu Du [1,2,3,4] ✉ & Weiping Su [1,2,3,4] ✉

Modular and regio-/stereoselective syntheses of all-carbon tetrasubstituted olefins from simple alkene materials remain a challenging project. Here, we demonstrate that a remote-carbonyl-directed palladium-catalyzed Heck/isomerization/C(sp²)−H arylation sequence enables unactivated 1,1-disubstituted alkenes to undergo stereoselective terminal diarylation with aryl iodides, thus offering a concise approach to construct stereodefined tetrasubstituted olefins in generally good yields under mild conditions; diverse carbonyl groups are allowed to act as directing groups, and various aryl groups can be introduced at the desired position simply by changing aryl iodides. The stereo-control of the protocol stems from the compatibility between the E/Z isomerization and the alkenyl C(sp²)−H arylation, where the vicinal group-directed C(sp²)−H arylation of the Z-type intermediate product thermodynamically drives the reversible E to Z isomerization. Besides, the carbonyl group not only promotes the Pd-catalyzed sequential transformations of unactivated alkenes by weak coordination, but also avoids byproducts caused by other possible β-H elimination.

All-carbon tetrasubstituted olefins extensively exist in natural products, leading drugs and functional organic materials[1,2]. They are also key precursors for various useful transformations such as hydrogenation, epoxidation, reductive Heck and other processes that generate contiguous highly substituted C(sp³) centers[3]. The importance of such tetrasubstituted olefins in both academia and industry has stimulated the enduring efforts towards exploiting the catalytic methods for their syntheses[4–12]. In this context, the transition metal-catalyzed difunctionalization of alkynes via carbometallation stands for the most efficient and modular route to synthesize fully substituted olefins[3,13–22], but still suffers from the difficulty in achieving regio-/stereocontrol when applied to electronically or sterically unbiased substrates. In light of this, the alternative and general methods for the regio-/stereoselective construction of all-carbon tetrasubstituted olefins remain highly went after.

Since the pioneering work of Hallberg[23,24], the chelation-assisted intermolecular Heck reaction has been prospering and stood out as a powerful strategy to get stereodefined, highly functionalized olefins[25]. Terminal alkenes bearing a removable heteroatom-tethered 2-pyridyl or 2-pyrimidyl have been developed by Itami and Yoshida[26,27] to implement stereoselective sequential Heck arylation, followed with cross-coupling reactions to produce triaryl- and

[1]State Key Laboratory of Structural Chemistry, Fujian Institute of Research on the Structure of Matter, Chinese Academy of Sciences, Fuzhou, Fujian, PR China. [2]Fujian Science & Technology Innovation Laboratory for Optoelectronic Information of China, Fuzhou, Fujian, PR China. [3]University of Chinese Academy of Sciences, Beijing, PR China. [4]College of Chemistry & Materials Science, Fujian Normal University, Fuzhou, PR China. ✉e-mail: duyu@fjirsm.ac.cn; wpsu@fjirsm.ac.cn

**Fig. 1 | Modular and regio-/stereoselective synthesis of all-carbon tetra-substituted olefins. a** Yoshida's chelation-assisted regio-/stereoselective synthesis of tetra-arylated alkenes. **b** Studer's sequential oxidative Heck arylation for the stereoselective synthesis of tetrasubstituted olefins. **c** Dong's modular and regioselective synthesis of tetrasubstituted olefins enabled by an alkenyl Catellani reaction. **d** This work: native carbonyl group directed stereospecific sequential arylation of alkenes via a pivotal *E/Z* isomerization. Pd, palladium. Ar, aryl. DIBAL-H, Diisobutylalumium Hydride. E, electrophile. Nu, nucleophile. DG, directing group.

tetraaryl-alkenes with defined double bond geometry (Fig. 1a). Later on, Studer et al.[28] presented a nitroxide-mediated sequential oxidative Heck arylation for the stereoselective synthesis of tetra-substituted olefins from easily available acrylates. Evidence proved that the directing group (DG) is not necessary for this sequential arylation (Fig. 1b). In another scenario, an alkenyl Catellani reaction has been utilized for modular and regioselective synthesis of all-carbon tetrasubstituted olefins by Dong[29] (Fig. 1c), the use of a modified norbornene was critical to suppressing the undesired cyclopropanation pathway. By now, great efforts have been made on chelation-assisted alkenyl C−H activation[30,31], but mainly focused on exploring the reactivity of specific sites in various substrates, especially the C−H alkenylation, little work has been reported on sequential alkenyl C−H functionalization, not to mention the modular synthesis of fully substituted olefins[32–35], which would be even more a hostage to the steric hindrance.

As a long-term interest in challenging the C−H functionalization of simple raw materials assisted by a native DG, we envisioned whether we can take advantage of both the flexibility of chelation-assisted Heck arylation and the excellent regio-/stereocontrol of directed alkenyl C−H activation to implement a multistep sequential arylation process, and furthermore, its application in modular synthesis of fully substituted olefins would be a good touchstone. However, in practice, the chelation effect given by a native DG is generally too weak to support a thorough multi-fold Heck arylation, in another hand such a native DG directed distal alkenyl C−H functionalization is also nontrivial. To make the assumption come true, some specific research ideas are designed as follows: (i) olefinic substrates bearing weakly coordinating DGs would provide a chelation effect to bring the metal in close proximity to the alkenyl, (ii) a flexible-alkyl-chain-tethered DG could promote the formation and dissociation of less rigid bidentate complexes (Fig. 1d, I & II) possessing high conformational degree of freedom, (iii) the weak

chelation effect may not direct C−H activation but rather stabilizes the alkenyl-metal complex and facilitates the subsequent transformation[36–38], (iv) the chelation effect may facilitate a reversible Pd(II)-catalyzed internal alkene *E/Z* isomerization[39] to provide one more potential reactive site (Fig. 1d, (*E*)-**3** & (*Z*)-**3**), (v) this platform would contribute to improving chemo-/regio-/stereo-selectivity under mild conditions as well, and vi) using a native group as the weakly coordinating auxiliary would be benefit for simplifying syntheses and further derivatization.

Here, we describe our research of carbonyl-directed stepwise double arylation of 1,1-disubstituted olefins, via a sequential Mizoroki-Heck, *E/Z* isomerization and C−H activation (3 steps in 2 pots), implementing the modular synthesis of stereodefined 1,1,2-triarylated all-carbon tetrasubstituted olefins (Fig. 1d). Some common carbonyl groups can direct the reaction, particularly the aldehyde function works well as a weakly coordinating DG and remains in final products. Research indicates that the first Heck reaction can furnish a terminal mono-arylated product (*E*)-**3** in high stereoselectivity as we expected. Bearing a weakly coordinating DG such as aldehyde, ketone or ester, alkene (*E*)-**3** is unable to take a second Heck arylation to give the target molecule **4** (Fig. 1d, Path b), or undergo C−H activation via a twisted *endo*-metallacyclic intermediate **III** to generate the β-*trans*-arylated product *iso*-**4** (Fig. 1d, Path c). As an alternative, a reversible Pd(II)-catalyzed isomerization from (*E*)-**3** to its (*Z*)-isomer and subsequent vicinal group-directed alkenyl C−H arylation works well to form the target molecule **4** in a stereoconvergent fashion (Fig. 1d, Path a). Pd(II)-catalyzed internal alkene isomerization has been recognized as a ubiquitous transformation for more than 60 years[40], recently its combination with asymmetric catalysis has already been exploited in stereoconvergent synthesis[41–44]; however, its latent capacity in affecting stereoselective alkenyl C(sp²)−H functionalization seems to be neglected, particularly the thermodynamically unfavorable *E* to *Z* isomerization was believed to have potential practical value[45].

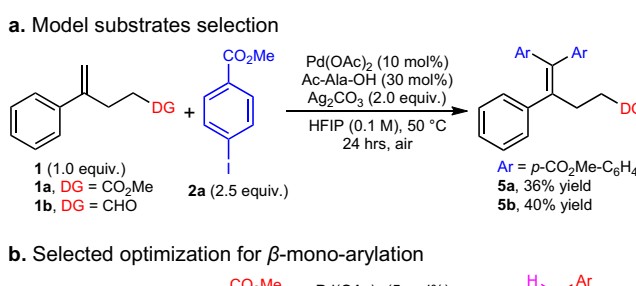

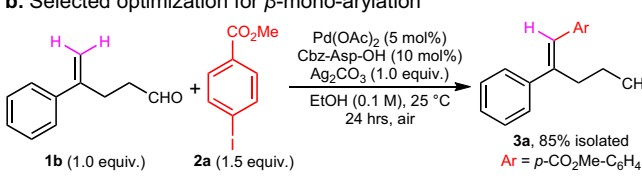

**Fig. 2 | Reaction condition optimization and substrate scope for β-mono-arylation. a** Model substrates selection. Reactions were run on a 0.2 mmol scale, yields are isolated yields. **b** Selected optimization for β-mono-arylation. Reactions were run on a 0.2 mmol scale, yields were determined by $^1$H-NMR spectroscopic analysis using 1,1,2,2-tetrachloroethane (0.1 mmol) as the internal standard, isolated yields are given in parentheses, E/Z ratios were determined by $^1$H-NMR spectroscopic analysis of the reaction crude. $^a$18% **5b**. $^b$20% **3a''**. **c** Substrate generality for β-mono-arylation. All data represent the average of (more than) two independent experiments, reactions were run on a 0.2 mmol scale, yields are isolated yields, E/Z ratios were determined by $^1$H-NMR spectroscopic analysis of the reaction crude. $^c$Gram-scale reaction. $^d$45 °C. $^e$Single-crystal structure of **3f**, determined by X-ray diffraction. w/o, without.

## Results and discussion

### Model substrates selection
Initially, methyl 4-phenylpent-4-enoate **1a** bearing a non-conjugated, weakly coordinating group was chosen as the model substrate. It has a pre-installed α-aryl group and is easily available (Supplementary Methods 1.2). Under palladium/mono-N-protected amino acid (MPAA)[46] cooperative catalysis, 1,1-disubstituted olefin **1a** could undergo β-diarylation with methyl 4-iodobenzoate **2a** to form the tetrasubstituted olefin **5a** in an isolated yield of 36% (Fig. 2a). During the screening of DGs under the same conditions, surprisingly, alkenyl aldehyde **1b** emerged as the best candidate to give the corresponding diarylated product **5b** in 40% yield (Fig. 2a). The aldehyde function has rarely been used as the actual DG in palladium catalysis, even in some latest reports on alkene functionalization, the aldehyde could only direct the reaction in case of the formation of the transient directing group[47–52] (TDG). Thus the olefin **1b** bearing a flexible-alkyl-chain-tethered aldehyde was chosen as the model substrate to test our hypothesis.

### β-Mono-arylation in EtOH
To explore the proposed sequential stereospecific arylation (Fig. 1d), the chelation-assisted β-mono-arylation of alkenyl aldehyde **1b** with methyl 4-iodobenzoate **2a** was first studied (Fig. 2b). After careful evaluation of various reaction parameters (Supplementary Methods 1.3.1), ultimately, in the presence of 5 mol% Pd(OAc)$_2$ and 10 mol%

N-Cbz-aspartic acid (Cbz-Asp-OH) as a ligand, and one equivalent of Ag$_2$CO$_3$ as the halide scavenger, after 24 h reaction at room temperature (25 °C) in ethanol under air, the desired mono-arylation product **3a** was isolated in 85% yield and approximately 8:1 cis/trans (E/Z) selectivity (Fig. 2b, entry 1). It was found that ethanol (or methanol) would give the highest yield and good E-selectivity when used as the solvent (entries 2–4). Control experiments indicate that palladium catalyst and silver salt are essential to this reaction, a moderate yield was observed in the absence of ligand, and increasing the temperature led to a diminished yield and selectivity (entries 5–8). When replacing the aldehyde function with a methyl group, without (w/o) the chelation effect, a diarylation **3a''** was isolated as the main product, indicating the importance of a DG in controlling the direction of β-H elimination (entry 9). Generally, MPAAs and limited phosphine-based ligands benefit the reaction (Supplementary Table 1).

With the optimized conditions for the chelation-assisted E-selective β-mono-arylation, substrate scope was investigated (Fig. 2c). In the reaction with alkenyl aldehyde **1b**, aryl iodides bearing para- or meta-electron withdrawing groups (e.g., ester, halogen, trifluoromethyl, nitro and cyano) afforded the desired products in good yields and stereoselectivities, electron donating groups (e.g., alkyl, alkyloxy) were also well tolerated (**3a** – **3n**). Note that reactions involving a functional group with coordinating property (e.g., nitro, cyano) need higher temperature (**3f** & **3g**). The derivative of (+)-Fenchol worked well to provide the corresponding product **3j**. The steric factors appeared to

be crucial for this reaction, for example, it is hard to get compound **3o** bearing *ortho*-fluorophenyl at room temperature, yet it could be obtained in 64% yield at a higher temperature (45 °C). Some bulky polycyclic aryl iodides (e.g., naphthyl) worked fine as well (**3p** & **3q**). Heteroaryl (e.g., benzofuryl, pyrrolyl, indolyl) coupling partners were not compatible, probably due to their relatively strong coordinating property. The absolute configuration of **3f** was determined by single-crystal X-ray diffraction, while other products were assigned based on NMR similarities.

The alkenyl substrate was then explored (Fig. 2c). Modifying the pre-installed phenyl group with an electron-withdrawing group or a large substituent led to an obviously decreased yield (**3r** & **3s**), illustrating the reduced electron density of the double bond is unfavorable for the reaction, and so does the increased steric hindrance close to the coordinating center. Adding one methylene unit to the alkyl chain furnished the *β*-mono-arylated product **3t** in a slightly decreased yield and higher selectivity. In these reactions, selectivities (*E/Z*) are all close to 10:1.

## Consecutive double arylation in HFIP

After introducing the first aryl by Heck reaction, we turned our attention to the second arylation to construct the target 1,1,2-triarylated olefins. As the consecutive diarylation in one pot was obtained in our preliminary study (Fig. 2a), we envisioned that the same condition might be suitable for the second arylation of compounds **3**. As a result, the consecutive diarylation of alkenyl aldehyde **1b** with methyl 4-iodobenzoate **2a** was studied (Fig. 3a).

After extensive research, the optimal conditions (catalyst, ligand, base, additive, etc.) have been determined eventually (Supplementary Methods 1.3.2). The desired diarylated product **5b** was isolated in 77% yield with 10 mol% Pd(OAc)$_2$ as the catalyst, 30 mol% *N*-Fmoc-phenylglycine (Fmoc-Phg-OH) as a ligand, 0.5 equivalent of aryl boronic acid as an additive and two equivalents of Ag$_2$CO$_3$ as the halide scavenger, after 24 h reaction at 45 °C in HFIP under air (Fig. 3a, entry 1). Hexafluoroisopropanol (HFIP) displays a unique solvent effect in facilitating diarylation[53] (entry 2). Control experiments show that palladium catalyst and silver salt are crucial to this reaction (entries 3 ~ 4). The absence of the aryl boronic acid caused a little decrease in the yield, meanwhile mono-arylated *α,β*-unsaturated aldehyde **6a** was obtained as a byproduct in less than 10% yield (entry 5). The aryl boronic acid could play an effect in slightly promoting the reaction in the absence of ligand (entries 6 ~ 7). In contrast to the obvious mono-arylation in ethanol without the ligand (Fig. 2b, entry 7), herein, there's almost no reaction in the absence of ligand and additive (entry 7). Replacing aryl iodide with aryl boronic acid led to the formation of mono-arylated product **3a** in only 23% yield (entry 8). Other sterically hindered aryl boronic acids, such as 2,6-dimethylphenyl boronic acid and 2,6-diisopropylphenyl boronic acid, were tested, the results were comparable with that obtained without using aryl boronic acids (Supplementary Table 16). These details reveal some important features of reaction mechanism (vide infra). Moreover, a comparable result was obtained at room temperature (25 °C, entry 9). Very little amount of the goal product could be observed when using non-chelating substrate (entry 10).

Under the optimized conditions, a variety of *para*- and *meta*-substituted aryl iodides could react with alkenyl aldehyde **1b** in one-pot to give the desired diarylation products in moderate to good yields (Fig. 3b), combined with a little mono-arylation. Both electron-poor and -rich aryl groups are tolerated (**5b** ~ **5s**), generally, the presence of an electron withdrawing group can promote the oxidative addition of aryl iodide to palladium, leading to a higher yield than that derived from an electron-rich analogue. Due to the steric hindrance, *ortho*-fluoro-aryl iodide could only generate the mono-arylated product in approximate 40% yield (**3o**). The derivative of anti-inflammatory and analgesic drug Ketoprofen was applicable in this reaction (**5m**). Some

multi-substituted aryl iodides were also tolerated, delivering the diarylation products in moderate to low yields (**5t** ~ **5v**). Similar to the *β*-mono-arylation (Fig. 2c), heteroaryl iodides were not compatible in this reaction, and even the cyano group.

Further investigation of alkenes was performed under optimal conditions with aryl iodide **2a** (Fig. 3b). In contrast to mono-arylated products **3r** and **3s** (Fig. 2c), it seems that steric factors appeared to be more dominant for diarylation (**5w** & **5x**) than electronic effects. For example, the *tert*-butyl substituted alkenyl substrate can only be diarylated in a low yield (**5x**), likely due to its aggravated distortion in structure. Some common native groups can also work as coordinating auxiliary to direct the diarylation (e.g., ester **5a**, ketone **5y**, carboxylic acid **5z**) except for the aldehyde. Replacing aldehyde with hydroxy led to a complicated result, such as arylation, isomerization, migration, oxidation, and etc, amide substrate failed to give any product and could be recovered almost completely, the substrate bearing a longer alkyl chain was not compatible either (Supplementary Fig. 1, 1c, f & g). The results suggest that weakly coordinating DGs and a proper length of alkyl chain are definitely important for the consecutive diarylation.

## Stepwise double arylation

Having established the efficient assembling methods for *β*-position mono- and di-arylation, we next examined the stepwise double arylation using different aryl iodides via the combination of these two approaches (Fig. 3c). Experimental results showed the strategy is feasible with a slight modification of optimal conditions. 1,1-Disubstituted olefin **1** was first subjected to the chelation-assisted *E*-selective *β*-mono-arylation in ethanol, after simple purification by filtrating over silica gel, the second *β*-arylation was furnished in HFIP to generate various stereodefined 1,1,2-triarylated tetrasubstituted olefins.

Noteworthy features are that (i) double arylation step-by-step with the same aryl iodide gives a comparable result with that derived from a one-pot diarylation in HFIP (e.g., **5h**), (ii) both electron-donating and -withdrawing group substituted aryl iodides are tolerated, however, an electron-donating group generally leads to low yield (**4m** & **4n**), (iii) the steric hindrance is the main factor constraining reactions (**4q**), the method is not applicable for bulky coupling partners (e.g., naphthyl, *tert*-butyl-aryl and multi-substituted aryl), (iv) changing the feeding sequence of aryl iodides could furnish the syntheses of *Z*- or *E*-type isomers separately (**4i** ~ **4l**), (v) the pre-installed *α*-aryl group can be varied as well (**4r** ~ **4t**), (vi) some native groups can also direct the stepwise double arylation to give the corresponding 1,1,2-triarylated olefins (e.g., ester **4u**, ketone **4v**), (vii) the absolute configuration of **4s** was determined by single-crystal X-ray diffraction, the configuration of **4r** was confirmed by analysis of NOE (Nuclear Overhauser Effect) interactions, and above all, (viii) the *E*-dominated mixed isomers **3** obtained in the first mono-arylation undergo a stereoconvergent transformation, an *E* to *Z* geometric isomerization combining with a subsequent arylation generates a single isomer **4**, any other possible isomers were not observed. Mechanistic studies proved the second arylation occurred in HFIP is not a Heck reaction (vide infra).

## Mechanistic studies

It is noteworthy that a poor *E/Z* ratio was obtained in isolated *β*-mono-arylation product in HFIP (Fig. 2b, entry 2; Fig. 3a, entry 8; Fig. 3b, **3o**). A series of control experiments were performed for mechanistic investigation (Fig. 4a-d, Supplementary Discussion 2.2). Parallel experiments of *β*-mono-arylation and *β*-diarylation of model substrates employing a smaller amount of aryl iodide **2a** were conducted separately (Fig. 4a). In contrast to the high *E/Z* selectivity observed in the *β*-mono-arylation (*E/Z* = 13:1), the mono-arylated product **3a** was also obtained in the *β*-diarylation experiment, albeit with a poor stereoselectivity (*E/Z* = 5:7), the latter was obviously distinguished from a normal Heck reaction which prefers to form the thermodynamically more stable *E* geometry. To gain insights into the reaction mechanism,

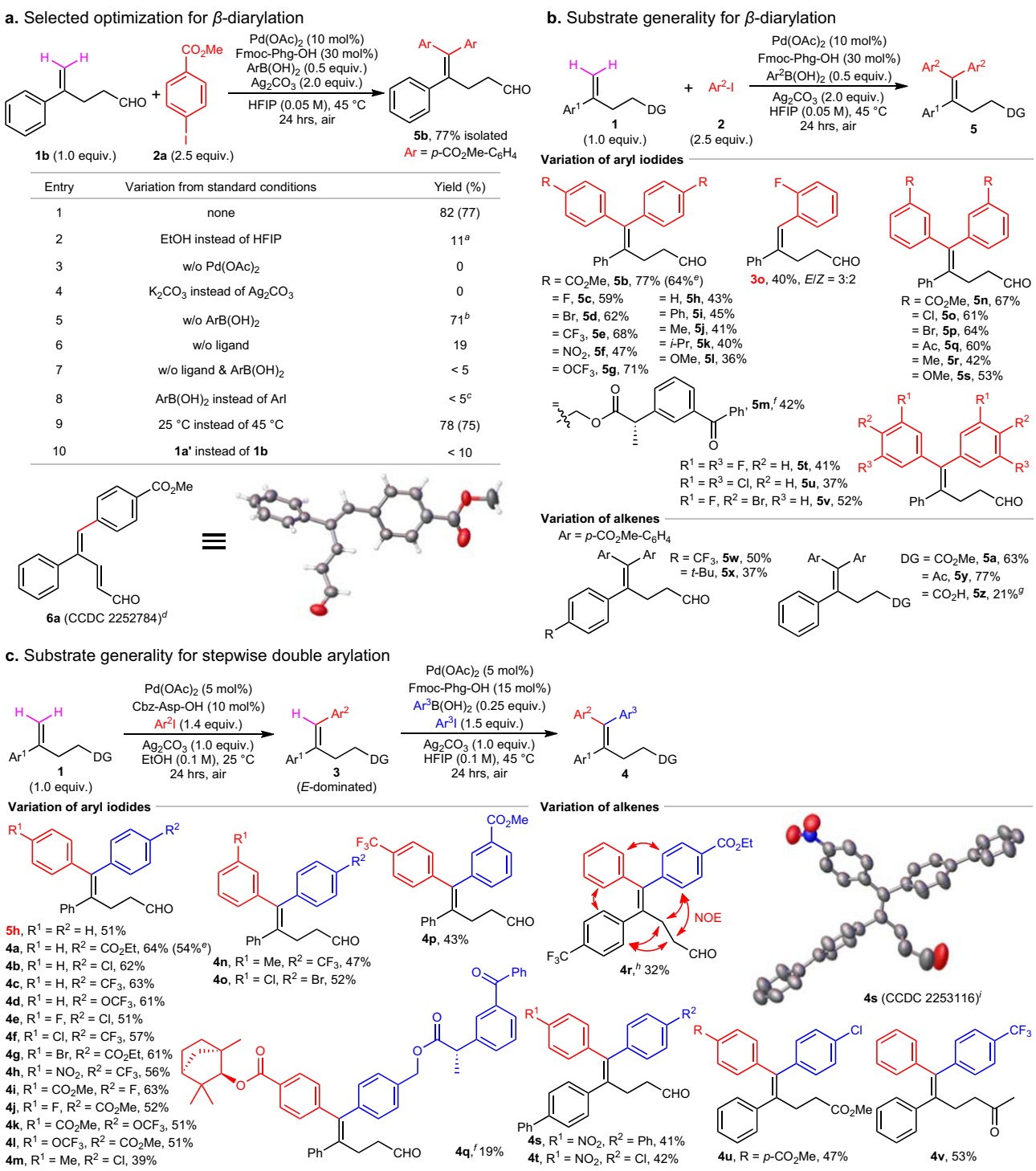

**Fig. 3 | Reaction condition optimization and substrate scope for consecutive and stepwise double arylation. a** Selected optimization for β-diarylation. Reactions were run on a 0.1 mmol scale, yields were determined by ¹H-NMR spectroscopic analysis using 1,1,2,2-tetrachloroethane (0.1 mmol) as the internal standard, isolated yields are given in parentheses. [a]43% **3a** (E/Z = 5:1, determined by ¹H-NMR spectroscopic analysis of the reaction crude). [b] < 10% **6a**. [c]23% **3a** (E/Z = 3:2). [d]Single-crystal structure of **6a**, determined by X-ray diffraction. **b** Substrate generality for β-diarylation. All data represent the average of (more than) two independent experiments, reactions were run on a 0.2 mmol scale, yields are isolated yields. [e]Gram-scale reaction. [f]without ArB(OH)₂. [g]Isolated yield after esterification with methyl iodide. **c** Substrate generality for stepwise double arylation. All data represent the average of (more than) two independent experiments, reactions were run on a 0.2 mmol scale, yields are isolated yields. [h]Confirmed by NOESY. [i]Single-crystal structure of **4s**, determined by X-ray diffraction. w/o, without. NOE, Nuclear Overhauser Effect.

the mono-arylated product **3b** (with an E/Z ratio of 10:1) obtained under the standard β-mono-arylation conditions was subjected to different control experiments without using aryl iodides, obvious E to Z isomerization was observed under palladium catalysis assisted by HFIP, aryl boronic acid, ligand and silver carbonate (Fig. 4b). And further, the mixed isomers **3b** (with an E/Z ratio of 5:6) could be transformed completely to furnish a single isomer **4w** under the modified β-diarylation conditions (Fig. 4c). The results look likely to be consistent

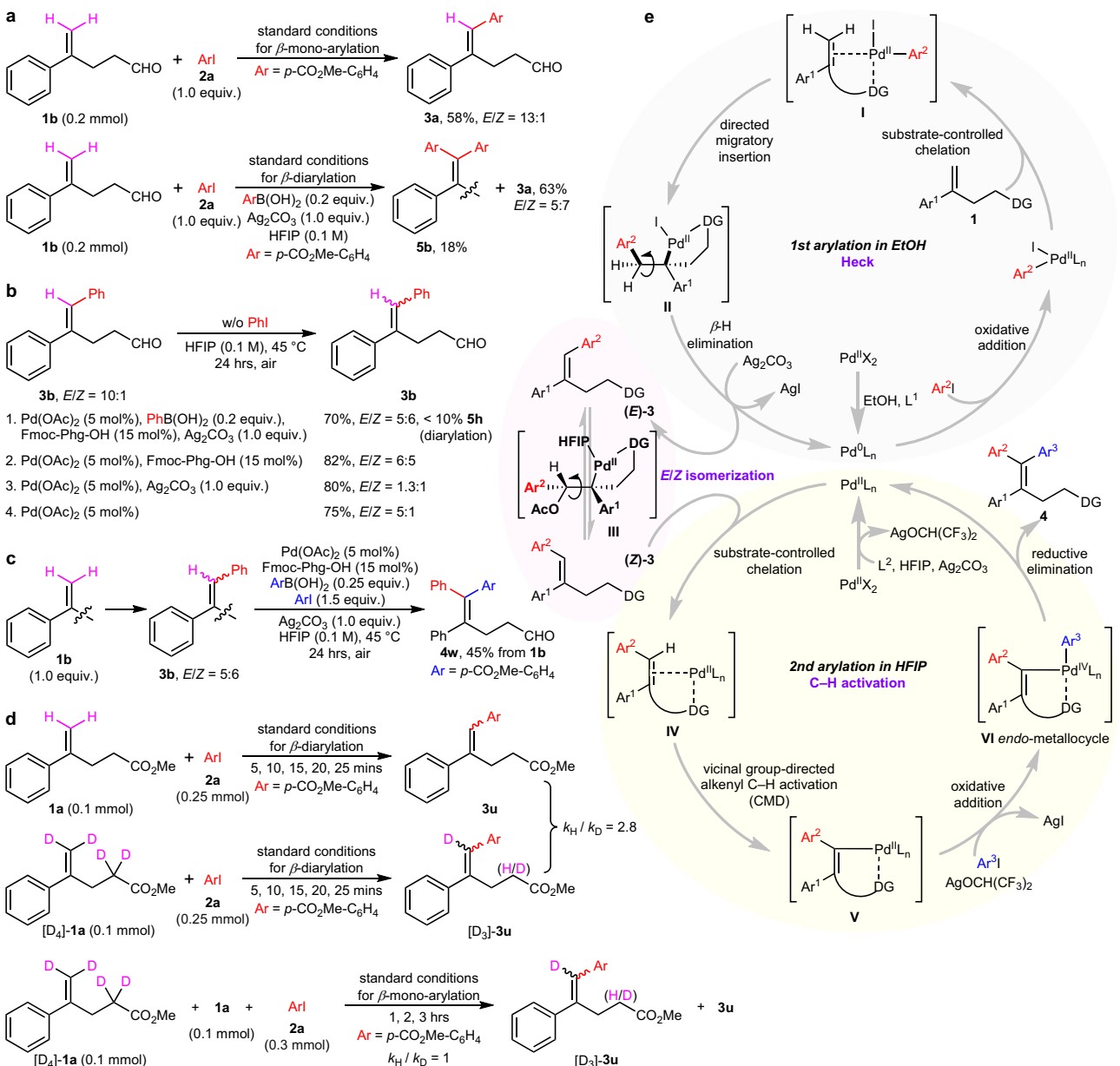

**Fig. 4 | Mechanistic studies. a** Parallel experiments of β-mono-arylation and β-diarylation of model substrates employing a smaller amount of aryl iodide **2a**, a poor stereoselectivity of mono-arylation was obtained in the latter case, suggesting that an *E/Z* isomerization is probably involved in the catalytic reaction in HFIP. **b** *E/Z* isomerization of the internal alkene **3b**. Obvious *E* to *Z* isomerization was observed under palladium acetate catalysis assisted by ligand or silver carbonate in HFIP. **c** The second arylation of the mixed isomers **3b** (*E/Z* = 5:6). The formation of the single isomer **4w** demonstrates that the second arylation in HFIP is a stereo-convergent procedure, probably consisting of a reversible Pd(II)-catalyzed *E* to *Z* isomerization of **3b** and subsequent vicinal group-directed alkenyl C–H arylation of (*Z*)-**3b**. **d** Deuterium KIE (kinetic isotope effect) experiments. A KIE data of $k_H/$

$k_D = 2.8$ was given in parallel experiments of **1a** and deuterated-**1a** with **2a** in HFIP, indicating that the C−H cleavage step might be the rate-determined step for the arylation occurred in HFIP; a value of $k_H/k_D = 1$ was obtained in intermolecular competitive KIE experiments of **1a** and deuterated-**1a** in ethanol, more in line with the characteristics of Heck reaction. **e** Outline of a possible pathway for the step-wise double arylation, some ligands and counter ions were omitted or simplified for clarity. A nucleopalladation via intermediate **III** is proposed as a key step to provide a stereoconvergent pathway, and a subsequent vicinal group-directed alkenyl C–H arylation via the CMD (concerted metalation deprotonation) mechanism collaborates well to provide the final product.

with alkene isomerization-then-C−H arylation. Parallel reactions of substrates **1a** and deuterated-**1a** with aryl iodide **2a** in HFIP were conducted (Fig. 4d), the KIE (kinetic isotope effect, $k_H/k_D = 2.8$) data obtained from their initial rate constants indicates that the C−H cleavage step is possible the rate-determined step for the arylation occurred in HFIP; in particular, double bond migration of **1a** was also observed during the reaction, suggesting that an allyl-Pd(II) intermediate is probably involved, nevertheless, terminal alkenyl C(sp²)−H arylation seems to remain dominant. By comparison, the

intermolecular competitive experiment in ethanol gave a value of $k_H/k_D = 1$ (Fig. 4d).

On the basis of experimental results and literature precedents, a plausible mechanism of this chelation-assisted stepwise double aryla-tion is proposed in Fig. 4e. Some ligands and counter ions were omitted for clarity. For the first β-mono-arylation in ethanol, a chelation-assisted intermolecular Heck reaction works smoothly to furnish the terminal arylated product **3** with a high *E/Z* selectivity. Theoretically, if the *E*-dominated mixture **3** still underwent the second

arylation in HFIP via a Heck-type pathway, the final stereoselectivity after β-H elimination would not be obviously better than that observed in the first arylation. But in fact, the final 1,1,2-triarylated olefin **4** was obtained as a single isomer. Interestingly, a reversible Pd(II)-catalyzed *E/Z* isomerization of *E*-dominated-**3** and subsequent vicinal group-directed alkenyl C–H arylation of (*Z*)-**3** take place in one pot, in a manner of dynamic kinetic resolution to promote the isomerization and the final formation of the single isomer **4**. Although the precise mechanism remains unclear at this stage, evidence suggests that both the directing group and the solvent HFIP display a critical role in alkene isomerization and subsequent C–H activation, thus an *anti*-nucleo-palladation-then-bond rotation-then-β-OAc elimination pathway with the formation of intermediate **III** could well explain the results[39].

For the alkenyl C(sp²)–H arylation that occurred in HFIP, the reactions preferably proceed via a Pd(II)/Pd(IV) catalysis under current conditions, particularly without using aryl boronic acids. What's more, the key *endo*-metallacyclic intermediates **V** and **VI** are nontrivial in vicinal group directed alkenyl C–H activation, there is a possible interconversion between *C*-enolate/oxa-π-allyl/*O*-enolate palladium intermediates or carbonyl-π-palladium that makes the metallocycle **V/VI** no larger than a 7-membered ring. The presence of aryl boronic acid possibly promotes the formation of active aryl-Pd(II) species, thereby circumventing the issue of side reactions (**6a** in Fig. 3a) and accelerating the alkenyl C(sp²)–H arylation. However, due to its reductivity exhibited in catalytic circle, the aryl boronic acid cannot replace the role of aryl iodides, even if exogenous oxidants were added. The consecutive double arylation performed in HFIP might occur via a C–H arylation-then-isomerization-then-C–H arylation pathway.

## Exploring optical properties

To further validate the potential use of synthesized tri-aryl all-carbon tetrasubstituted olefins in photoluminescence, we next investigated optical properties of compound **4s** which is less soluble in common solvents. As depicted in Fig. 5a, b, the fluorescence microscopy images

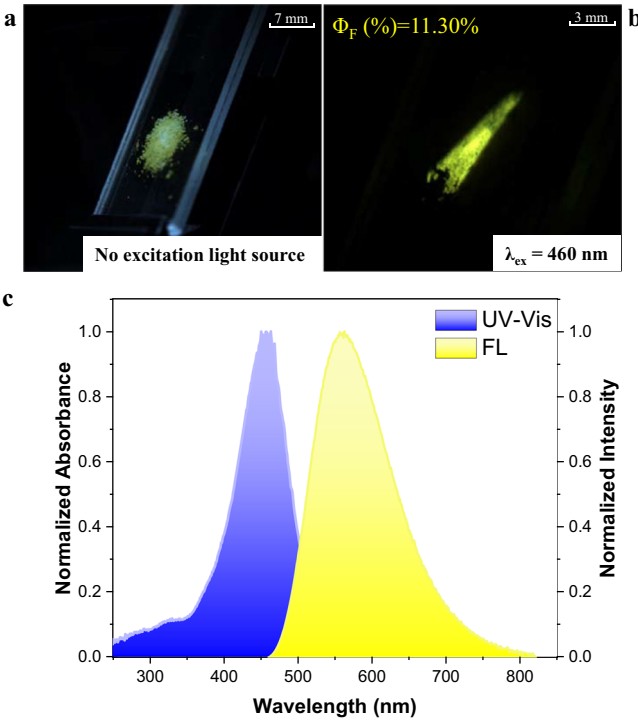

**Fig. 5 | Optical properties of compound 4s. a** Fluorescence image of the powder for compound **4s** without light source. **b** Fluorescence image of the powder for compound **4s** under excitation light (λ_ex = 460 nm). **c** Solid-state UV/Vis absorption (UV/Vis) and fluorescence emission (FL) spectra of **4s**.

show that solid-state **4s** emits strong yellow fluorescence with quantum yield about 11.3% under the excitation of light (λ_ex = 460 nm). The solid-state **4s** exhibits a broad absorption band in the visible region of 350–550 nm (with a maximum λ_ab = 450 nm), and its fluorescence emission spectrum is centered at 560 nm (Fig. 5c). Compound **4s** displays a large Stokes shift of 110 nm, which indicates the exciton self-quenching could be suppressed effectively.

In summary, a remote-carbonyl-directed palladium-catalyzed sequential Heck/isomerization/C(sp²)–H arylation of 1,1-disubstituted olefins with aryl iodides has been established, various stereodefined 1,1,2-triarylated all-carbon tetrasubstituted olefins were prepared in a modular and stereoconvergent fashion. Several attractive features of this synthetic strategy should be noteworthy. First, all components assembled stem from readily available or easily prepared raw materials. Second, this protocol features mild reaction conditions, operationally simple procedures and diversified product structures. Third, various aryls can be introduced at the desired position simply by changing the coupling partners and adding in an appropriate order, a simple alteration of addition order results in producing possible isomers. Fourth, such a stereoconvergent, remote-chelation-assisted alkenyl C–H arylation open up a door for distal alkenyl C–H functionalization. Fifth, the use of weakly coordinating DGs, especially the direct use of aldehyde function, provides more possibility to achieve diversified transformations in an atom- and step-economic manner. Although some compatibility issues still exist (e.g., heteroaryl iodides), this strategy provides a highly flexible platform to access diverse tri-aryl all-carbon tetrasubstituted olefins which would find many uses in the development of functional organic materials and pharmaceuticals. Further efforts will focus on the investigation into other kinds of functionalization as well as expanding the applicable coupling reagents.

## Methods

### General procedure for β-mono-arylation

To an oven-dried 35 mL Schlenk tube with previously placed magnetic stir-bar were added aryl iodide **2** (0.3 mmol, 1.5 equiv.), Pd(OAc)₂ (2.2 mg, 0.01 mmol, 5 mol%), Cbz-Asp-OH (5.3 mg, 0.02 mmol, 10 mol%), Ag₂CO₃ (55 mg, 0.2 mmol, 1 equiv.), followed by addition of EtOH (2 mL) and alkene substrate **1** (0.2 mmol). The tube was sealed with a screw cap and the reaction mixture was stirred vigorously at room temperature (25 °C). After stirring for 24 h, the resultant solution was filtered through a short pad of 1:1 mixture of Celite and silica gel, and the column was washed with ethyl acetate (15 mL). The combined organic solutions were concentrated under reduced pressure, and the residue was purified by flash column chromatography on silica gel to afford the desired product **3**.

### General procedure for consecutive double arylation

To an oven-dried 35 mL Schlenk tube with previously placed magnetic stir-bar were added aryl iodide **2** (0.5 mmol, 2.5 equiv.), aryl boronic acid (0.1 mmol, 0.5 equiv.), Pd(OAc)₂ (4.5 mg, 0.02 mmol, 10 mol%), Fmoc-Phg-OH (22.4 mg, 0.06 mmol, 30 mol%), Ag₂CO₃ (110 mg, 0.4 mmol, 2 equiv.), followed by addition of HFIP (4 mL) and alkene substrate **1** (0.2 mmol). The tube was sealed with a screw cap and the reaction mixture was stirred vigorously on a hotplate at 45 °C for 24 h. After completion of the reaction, the resultant solution was filtered through a short pad of 1:1 mixture of Celite and silica gel, and the column was washed with ethyl acetate (15 mL). The combined organic solutions were concentrated under reduced pressure, and the residue was purified by flash column chromatography on silica gel to afford the desired product **5**.

### General procedure for stepwise double arylation

To an oven-dried 35 mL Schlenk tube with previously placed magnetic stir-bar were added aryl iodide (0.28 mmol, 1.4 equiv.), Pd(OAc)₂

(2.2 mg, 0.01 mmol, 5 mol%), Cbz-Asp-OH (5.3 mg, 0.02 mmol, 10 mol%), Ag$_2$CO$_3$ (55 mg, 0.2 mmol, 1 equiv.), followed by addition of EtOH (2 mL) and alkene substrate **1** (0.2 mmol). The tube was sealed with a screw cap and the reaction mixture was stirred vigorously at room temperature (25 °C). After stirring for 24 h, the resultant solution was filtered through a short pad of 1:1 mixture of Celite and silica gel, and the column was washed with ethyl acetate (15 mL). The combined organic solutions were concentrated under reduced pressure to afford the mono-arylated product **3** which was used in the next step without further purification. To another oven-dried 35 mL Schlenk tube with previously placed magnetic stir-bar were added the second aryl iodide (0.3 mmol, 1.5 equiv.), aryl boronic acid (0.05 mmol, 0.25 equiv.), Pd(OAc)$_2$ (2.2 mg, 0.01 mmol, 5 mol%), Fmoc-Phg-OH (11.2 mg, 0.03 mmol, 15 mol%), Ag$_2$CO$_3$ (55 mg, 0.2 mmol, 1 equiv.), followed by the addition of the crude mono-arylated product **3** dissolved in 2 ml HFIP. The tube was sealed with a screw cap and the reaction mixture was stirred vigorously on a hotplate at 45 °C for 24 h. After completion of the reaction, the resultant solution was filtered through a short pad of 1:1 mixture of Celite and silica gel, and the column was washed with ethyl acetate (15 mL). The combined organic solutions were concentrated under reduced pressure, and the residue was purified by flash column chromatography on silica gel to afford the desired double arylated product **4**.

## Data availability
Crystallographic data for the structures reported in this Article have been deposited at the Cambridge Crystallographic Data Centre (CCDC), under deposition numbers CCDC 2252784 (**6a**), 2252796 (**3f**) and 2253116 (**4s**). Copies of the data can be obtained free of charge via http://www.ccdc.cam.ac.uk/data_request/cif. Experimental procedures, characterization of new compounds and all other data supporting the findings are available in the Supplementary Information. Source data of photophysical properties (**4s**) are provided with this paper. All data are available from the corresponding author upon request. Source data are provided with this paper.

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

## Acknowledgements

This work was financially supported by the National Key Research and Development Program of China (2018YFA0704502, W.S.), the National Natural Science Foundation of China (21871261, Y.D. & 21931011, W.S.), and Fujian Science & Technology Innovation Laboratory for Optoelectronic Information of China (2021ZZ105, W.S.).

## Author contributions

Y.D. and W.S. conceived the research concept. W.S. directed the project. R.L. and Y.D. designed and performed experimental studies. P.L. helped with the collection of crystallographic data of some new compounds and data analysis. Y.D. wrote the manuscript. K.L. participated in complementing experiments. All authors discussed the results, reviewed and edited the manuscript.

## Competing interests

The authors declare no competing interests.
