## [Peer Review File · Nature Communications]

REVIEWER COMMENTS

Reviewer #1 (Remarks to the Author):

Tetrasubstituted alkenes represent one of the most important classes of organic compounds due to their widely occurring in natural products, pharmaceuticals, and functional materials. The synthesis of these compounds in a regio- and stereoselective manner has provided major challenges to synthetic organic chemists due to the congested nature of the double bond. Su, Du, and coworkers have demonstrated that all-carbon tetrasubstituted alkenes could be obtained via a palladium-catalyzed Heck/E/Z isomerization/C(sp²)-H arylation cascade reaction. Preliminary mechanistic studies showed that an E/Z isomerization of mono-arylated product was occurred using HFIP as solvent, and the mixture of E/Z isomers of mono-arylated product could converted to the di-arylated product in high stereoselectivity.

If the second arylation reaction was also proceeded via Heck reaction, the di-arylated product would be more likely formed in an E/Z mixture. Thus the author proposed a C-H activation pathway for the second arylation reaction, which seems reasonable. However, more control experiments should be conducted such as deuteration and KIE experiments.

Although this manuscript still has some limitations including low to moderate yields and the requirement of aryl boronic acid (with the same aryl group of the corresponding aryl iodides) as additive, it is a nice piece of work which may stimulate the development of alkenyl C-H activations using weakly directing groups.

Overall, the referee would like to recommend the acceptance of this manuscript with major revision.

1) Other sterically hindered aryl boronic acids, such as 2,6-dimethylphenyl boronic acid, are encouraged to test. Maybe, they could promote the reactions without reacted with the substrates.

2) "Ar = p-CO₂Me-Ph" should be "Ar = p-CO₂Me-C₆H₄"

Reviewer #2 (Remarks to the Author):

The manuscript reported the study of a remote-carbonyl-directed palladium-catalyzed Heck/E/Z isomerization/C(sp²)-H arylation of alkenes, which leads to the syntheses of stereodefined tetrasubstituted olefins. Diverse carbonyl groups such as aldehyde, ketone, ester or carboxylic acid, can be used as native directing groups. The reaction operates under mild conditions (rt or 45 oC) and gives generally good yields. The reaction involves a sequential Heck/E/Z isomerization/C(sp²)-H arylation process, which was supported by the mechanistic studies, and a reasonable pathway was proposed. It is recommended for publication in Nature Communications after address the following minor revisions and questions before publication:

1. In the context of "Replacing the pre-installed aryl group with an electron deficient moiety or a large substituent led to an obviously decreased yield (3r & 3s)", it is not replacing the aryl group, but is varying the substituents on the aryl group. The statement should be revised.

2. Only arylation of alkenes were conducted in the study using ArI, how do alkenyl, alkynyl or alkyl halides perform in the reaction?
3. For the proposed mechanism, is Pd(II)/Pd(IV) catalysis also possible for the C(sp²)-H arylation step?

Many thanks to reviewers for her/his instructive comments and constructive suggestions, that are very helpful for improving the quality of the manuscript. In response to these suggestions, more experiments have been done and the results obtained from these experiments have been added to the revised manuscript. The following are our point-by-point response to the reviewers' comments.

Reviewer #1 (Remarks to the Author):

Tetrasubstituted alkenes represent one of the most important classes of organic compounds due to their widely occurring in natural products, pharmaceuticals, and functional materials. The synthesis of these compounds in a regio- and stereoselective manner has provided major challenges to synthetic organic chemists due to the congested nature of the double bond. Su, Du, and coworkers have demonstrated that all-carbon tetrasubstituted alkenes could be obtained via a palladium-catalyzed Heck/*E/Z* isomerization/ $C(sp^2)$ -H arylation cascade reaction. Preliminary mechanistic studies showed that an *E/Z* isomerization of mono-arylated product was occurred using HFIP as solvent, and the mixture of *E/Z* isomers of mono-arylated product could converted to the di-arylated product in high stereoselectivity.

If the second arylation reaction was also proceeded via Heck reaction, the di-arylated product would be more likely formed in an *E/Z* mixture. Thus the author proposed a C-H activation pathway for the second arylation reaction, which seems reasonable. However, more control experiments should be conducted such as deuteration and KIE experiments.

Although this manuscript still has some limitations including low to moderate yields and the requirement of aryl boronic acid (with the same aryl group of the corresponding aryl iodides) as additive, it is a nice piece of work which may stimulate the development of alkenyl C-H activations using weakly directing groups.

Overall, the referee would like to recommend the acceptance of this manuscript with major revision.

Response: In order to address the reviewer's concern about the precise mechanism of the second arylation process, deuterium kinetic isotope effect studies were performed. As shown in the revised manuscript (Fig. 4d) and Supplementary Information (Section 5.4), initial rate constants were measured by NMR spectroscopy and the relative ratio of k_H/k_D was obtained. As shown below (eq. 1), at first, we

planned to prepare deuterated-**3u** and use it in parallel controlled experiments for KIE studies. However, the deuterated ratio at β -position decreased significantly during the mono-arylation, and an inseparable byproduct **3u'** was observed in an approximate yield of 12%, which mainly due to the double bond migration and would seriously interfere with the following KIE studies. Although the decline in deuterated ratio can be suppressed by using deuterated methanol as the solvent, but the byproduct **3u'** is still exist. As a result, **[D₄]-1a** was used directly to test its initial rate constant of mono-arylation in HFIP (eq. 2), which compared with that obtained from the reaction of substrate **1a** (eq. 3); the KIE data ($k_H/k_D = 2.8$) shown that the C–H cleavage step might be the rate-determined step for the arylation occurred in HFIP. By comparison, the intermolecular competitive experiment in ethanol gave a value of $k_H/k_D = 1$ (eq. 4). The results can sever as a good evidence to support our proposed mechanism, in which the mono-arylation or diarylation occurred in HFIP are very likely to undergo a C(sp²)–H activation.

1. Other sterically hindered aryl boronic acids, such as 2,6-dimethylphenyl boronic acid, are encouraged to test. Maybe, they could promote the reactions without reacted with the substrates.

Response: We really appreciate the reviewer giving this suggestion. Some

sterically hindered aryl boronic acid, such as 2,6-dimethylphenyl boronic acid and 2,6-diisopropylphenyl boronic acid, have been tested in both consecutive diarylation (Supplementary Table 16) and stepwise double arylation; however, it seemed that they did not influence the reactions, comparable results were given as that obtained in the absence of aryl boronic acids.

2. “Ar = *p*-CO₂Me-Ph” should be “Ar = *p*-CO₂Me-C₆H₄”.

Response: We have changed “Ar = *p*-CO₂Me-Ph” to “Ar = *p*-CO₂Me-C₆H₄” in revised figures and Supplementary Information.

Reviewer #2 (Remarks to the Author):

The manuscript reported the study of a remote-carbonyl-directed palladium-catalyzed Heck/*E/Z* isomerization/*C*(sp²)-H arylation of alkenes, which leads to the syntheses of stereodefined tetrasubstituted olefins. Diverse carbonyl groups such as aldehyde, ketone, ester or carboxylic acid, can be used as native directing groups. The reaction operates under mild conditions (rt or 45 °C) and gives generally good yields. The reaction involves a sequential Heck/*E/Z* isomerization/*C*(sp²)-H arylation process, which was supported by the mechanistic studies, and a reasonable pathway was proposed. It is recommended for publication in Nature Communications after address the following minor revisions and questions before publication:

1. In the context of “Replacing the pre-installed aryl group with an electron deficient moiety or a large substituent led to an obviously decreased yield (**3r** & **3s**)”, it is not replacing the aryl group, but is varying the substituents on the aryl group. The statement should be revised.

Response: The statement has been revised as “Modifying the pre-installed phenyl group with an electron-withdrawing group or a large substituent led to an obviously decreased yield (**3r** & **3s**)”.

2. Only arylation of alkenes were conducted in the study using ArI, how do alkenyl, alkynyl or alkyl halides perform in the reaction?

Response: We appreciate the reviewer giving this suggestion. Indeed, starting

from 4-phenylpent-4-enal **1b** and mono-arylated product **3b**, we have already tried some common conditions (as shown below) for alkenyl C(sp²)-H alkenylation, alkynylation, alkylation and so on. However, starting materials were recovered in most cases; the existing reaction systems are not suitable for other functionalization besides arylation. As a result, our group is still working on exploiting other kinds of alkenyl C-H functionalization based on a sequential Heck/*E/Z* isomerization/C(sp²)-H activation process.

3. For the proposed mechanism, is Pd(II)/Pd(IV) catalysis also possible for the C(sp²)-H arylation step?

Response: Many thanks to the reviewer for her/his timely reminder. We focused on exploring the role of aryl boronic acid displayed in the reaction, but neglected the arylation reactions without these additives. In fact, under the existing conditions, the arylation reactions occurred in HFIP preferably proceed via a Pd(II)/Pd(IV) catalysis. The related discussion has been revised in the manuscript and a modified possible mechanism was given (Fig. 4e).

REVIEWERS' COMMENTS

Reviewer #1 (Remarks to the Author):

The authors have adequately addressed my previous concerns, and the revised manuscript has been intensively enriched. I support its publication in the esteemed Nat. Commun.